# Mediodorsal thalamus is required for discrete phases of goal-directed behavior in macaques

Evan Wicker[1], Janita Turchi[2], Ludise Malkova[1,3], Patrick A Forcelli[1,3,4]*

[1]Department of Pharmacology and Physiology, Georgetown University, Washington, United States; [2]Laboratory of Neuropsychology, National Institute of Mental Health, Maryland, United States; [3]Interdisciplinary Program in Neuroscience, Georgetown University, Washington, United States; [4]Department of Neuroscience, Georgetown University, Washington, United States

**Abstract** Reward contingencies are dynamic: outcomes that were valued at one point may subsequently lose value. Action selection in the face of dynamic reward associations requires several cognitive processes: registering a change in value of the primary reinforcer, adjusting the value of secondary reinforcers to reflect the new value of the primary reinforcer, and guiding action selection to optimal choices. Flexible responding has been evaluated extensively using reinforcer devaluation tasks. Performance on this task relies upon amygdala, Areas 11 and 13 of orbitofrontal cortex (OFC), and mediodorsal thalamus (MD). Differential contributions of amygdala and Areas 11 and 13 of OFC to specific sub-processes have been established, but the role of MD in these sub-processes is unknown. Pharmacological inactivation of the macaque MD during specific phases of this task revealed that MD is required for reward valuation and action selection. This profile is unique, differing from both amygdala and subregions of the OFC.
DOI: https://doi.org/10.7554/eLife.37325.001

*For correspondence:
paf22@georgetown.edu

**Competing interests:** The authors declare that no competing interests exist.

## Introduction

In daily life, reward contingencies are often unstable; an action that once produced a valued outcome may over time become less desirable. The ability to shift responses away from the previously valued action-outcome pair is a hallmark example of behavioral flexibility. Flexible responding (i.e., adapting behavior to reflect new reward contingencies or reward values) has been evaluated extensively through the use of reinforcer devaluation tasks (*Málková et al., 1997*; *Hatfield et al., 1996*).

In these tasks, the value of secondary reinforcers or operanda (e.g., objects), and actions that were once favored decrease following an experimental reduction in the value of the associated primary reinforcer. Processes required for optimal performance in the reinforcer devaluation task include: [1] registering a change in value of the primary reinforcer, [2] integrating the new value of the primary reinforcer with the cognitive representation of the secondary reinforcers/operanda (e.g., objects that signal the reward), and [3] guiding action selection to target the now optimal choice.

In the standard version of this task used in macaques, animals are trained to associate sets of objects with one of two rewards (primary reinforcers; e.g., peanuts or fruit snacks). After the association between objects and particular rewards is established, subjects are presented with a forced choice between objects associated with each of the two foods. The proportion of choices between the objects rewarded with one type of food (e.g., peanuts) versus the other (e.g., fruit snacks) represents the baseline preference. An experimental reduction of reward value by selective satiation (i.e., providing one food to satiety) produces a devaluation effect, i.e., a decrease in the proportion of objects associated with the sated food that are selected.

**eLife digest** Most of us have experienced feeling full after a main course, only to discover that we somehow still have room for dessert. Eating a particular foodstuff to the point of satiety makes that item temporarily less appealing. This is an example of reward devaluation. We typically respond to this phenomenon by adjusting our behavior. We give up on the main course, for example, and turn our attention instead to dessert. This ability to adjust our actions based on changes in the value of their outcomes is a form of behavioral flexibility.

Several brain regions contribute to behavioral flexibility. These include the amygdala, parts of the orbitofrontal cortex, and the mediodorsal thalamus. Wicker et al. have now explored the role of the mediodorsal thalamus by temporarily inactivating it in monkeys performing a task involving reward devaluation. The monkeys learned to associate one set of objects with peanuts and another with fruit. They were then given unlimited access to either peanuts or fruit. Finally, they were offered a choice between the two sets of objects. Like people who opt for dessert rather than another helping of a main course, the monkeys that had received peanuts chose the objects associated with fruit, and vice versa.

Temporarily inactivating the mediodorsal thalamus prevented this change in behavior. This occurred if the inactivation took place while the monkeys had unlimited access to the reward, or if it took place while they were choosing between the two objects. The mediodorsal thalamus is thus required both to update the value of a reward and to select the best course of action. This is in contrast to the amygdala and the orbitofrontal cortex, which each support only one of these processes.

Impaired behavioral flexibility is a hallmark of neuropsychiatric disorders, including addiction. Understanding the brain networks that support flexible responding may help improve the treatment of such disorders. As therapies that involve electrically stimulating the brain become more common, knowing which regions to avoid will be just as important as identifying new targets.

DOI: https://doi.org/10.7554/eLife.37325.002

These processes critically rely on an interactive network including the orbitofrontal cortex (OFC), the amygdala, and the mediodorsal thalamus (MD). The amygdala projects directly to OFC, and indirectly to the OFC via the MD (*Timbie and Barbas, 2015*). MD is reciprocally interconnected with two critical subregions of the OFC, Areas 11 and 13 (*Ray and Price, 1993*). Lesions to each of these nodes impair the typical shift away from the objects that predict the sated food (*Málková et al., 1997*; *Izquierdo et al., 2004*; *Izquierdo and Murray, 2010*; *Browning et al., 2015*; *Pickens, 2008*; *Mitchell et al., 2007*). Moreover, crossed lesions of any two nodes of this circuit likewise impair performance. While lesions provide strong evidence that these brain regions are required for task performance, they do not have the temporal specificity needed to dissociate the contributions during discrete phases of task performance. By contrast, focal pharmacological manipulations, which can transiently suppress activity within a brain region, have revealed differential roles for the amygdala and the OFC in particular phases of this task (*Wellman et al., 2005*; *West et al., 2011*). While neither the amygdala, nor the OFC are needed to register a change in value of the primary reinforcer, both structures play critical and differential roles in the subsequent processes. The amygdala is necessary for adjusting the object representations to reflect the new value of the primary reinforcer, but not necessary for optimal action selection (*Wellman et al., 2005*); a similar profile has also been observed for the OFC Area 13. By contrast, Area 11 is critical only for action selection (*Murray et al., 2015*).

Because MD receives input from amygdala and is reciprocally connected with both Areas 11 and 13 of the OFC (*Timbie and Barbas, 2015*), it suggests that this structure may be central to the devaluation circuit. However, the role of the MD in specific phases of devaluation is unknown. Here, we considered two competing hypotheses regarding the role of the MD in this network: [1] it mirrors the function of the amygdala and Area 13 of the OFC, serving primarily as a 'relay' to process information regarding value updating between the amygdala, Area 11 and Area 13, and [2] it mirrors the function of Area 11 of the OFC, contributing primarily to action selection. To dissociate between

these outcomes, we tested four adult male rhesus macaques on a reinforcer devaluation task while transiently inactivating the MD during various stages of the task (*Málková et al., 1997*).

## Results

Animals were first trained on a set of forty concurrent object discriminations; through repeated trials the animals learned the association between each object and a particular reward (see Materials and methods). Next, they were tested for preference on a baseline object probe test (baseline probe). In this test, two objects, each baited with one of the two foods, were pitted against each other. On another day, animals underwent selective satiation (pre-feeding with one of the two foods) and were again tested on an object probe test (sated probe), administered in a similar way as the baseline probe. This weekly sequence (baseline probe followed by sated probe) was repeated such that each food was sated for each experimental condition (see *Figure 1* for experimental timeline).

In the above probe tests, the animal selected between two objects, each baited with one of the two food rewards. Thus, the objects were the cue used to guide action. We also tested animals in a 'consummatory' probe, in which they chose between two competing food rewards in the absence of objects; this served as a control to ensure successful devaluation of the primary reinforcer.

A shift in choices between the baseline and sated probe tests is reflected by a positive proportion shifted (see Materials and methods) and indicates successful devaluation. A value of 1 indicates a complete shift away from objects predicting the sated food, a value of 0 indicates no shift in preference. Thus, decreases in proportion shifted after experimental manipulations indicate impaired reinforcer devaluation.

On separate testing weeks, we microinjected the $GABA_A$ receptor agonist muscimol (MUS, 9 nmol), the glutamate receptor antagonist kynurenic acid (KYNA, 450 nmol), or saline into the MD, either prior to selective satiation or prior to the sated probe test (*Figures 2A1–4*). MRI-guided stereotaxic targeting of the MD was performed and confirmed (*Figure 2B*) as we have described for

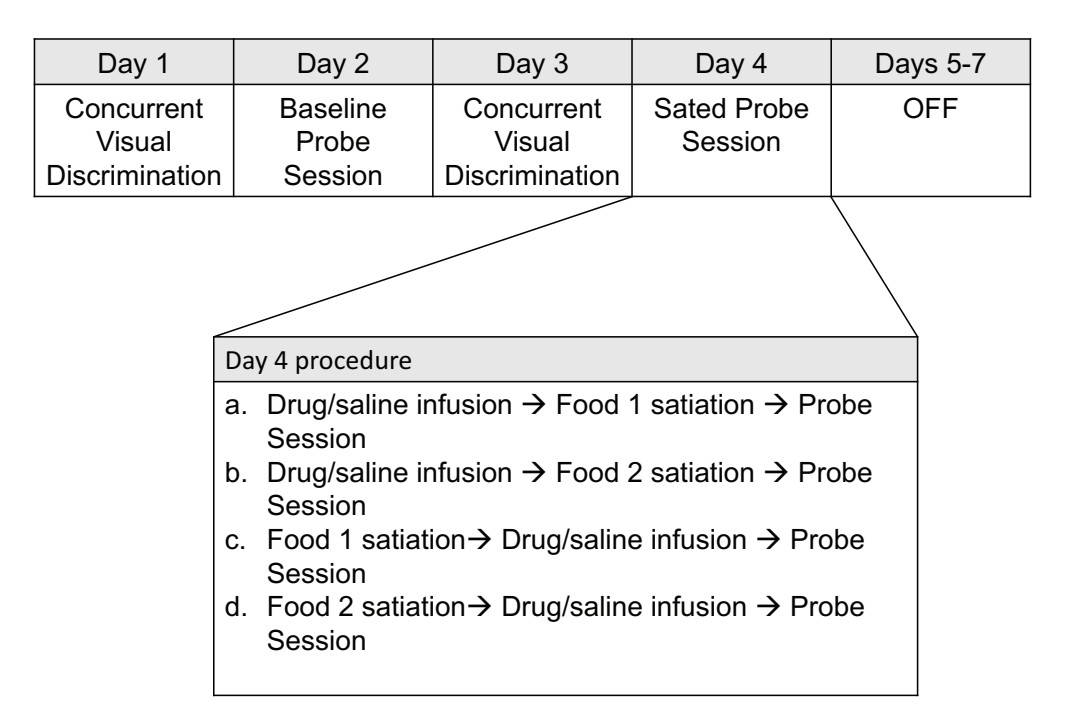

**Figure 1.** Weekly schedule of testing sessions. Days 1–7 represent a sequence of daily behavioral training. Testing order was pseudorandomized for each animal on the infusion probe sessions conducted on Day 4.
DOI: https://doi.org/10.7554/eLife.37325.003

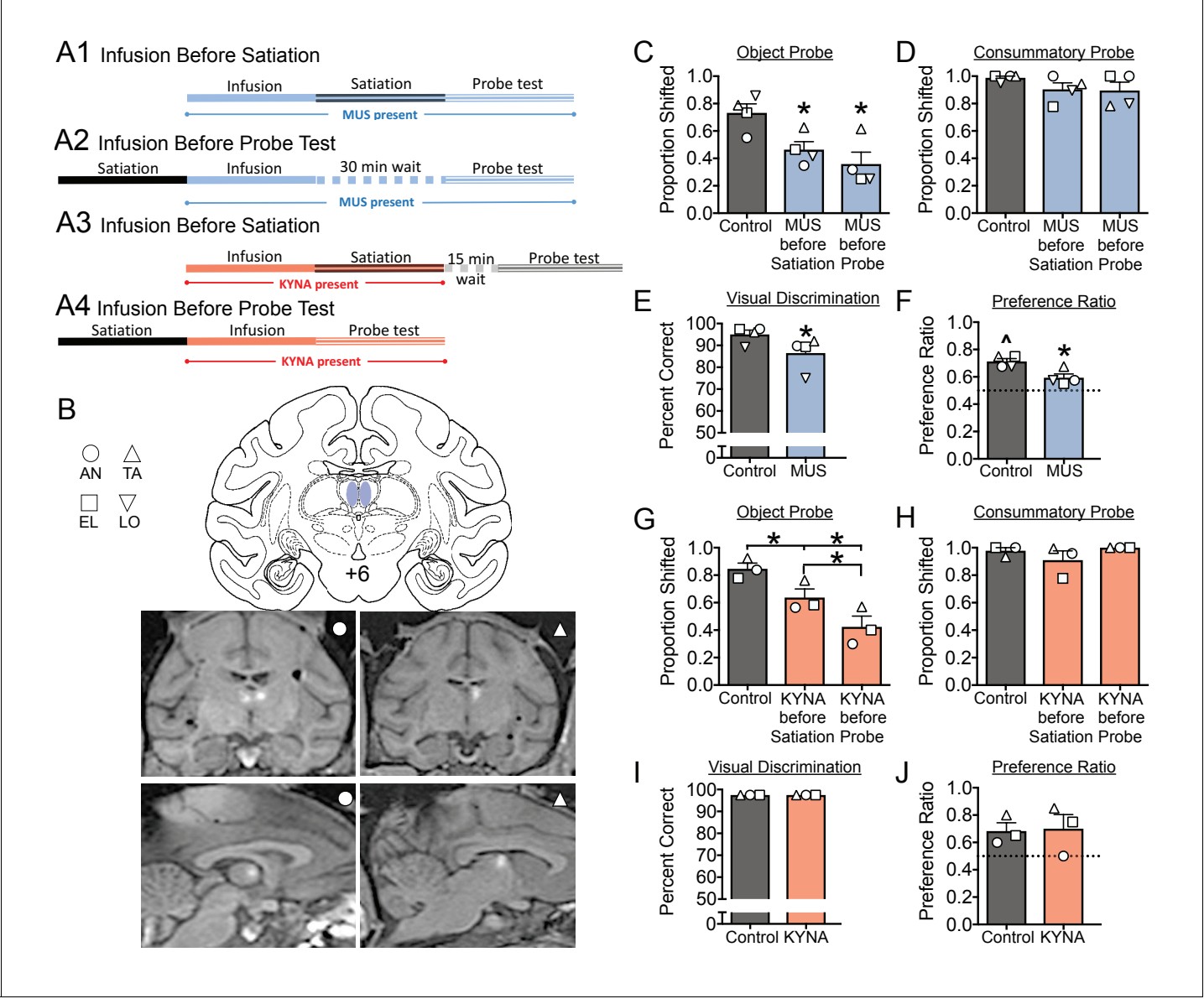

**Figure 2.** (A) Schematic, indicating the timing of drug infusions and tests. Muscimol (MUS, blue), was infused either before satiation (A1), and was thus present during both satiation and the probe test, or infused 30 min before the probe test, and was thus present only during the probe test (A1). The 30 min interval between infusion and test in (A2) was selected to match the interval between infusion and probe in (A1). Kynurenic acid (KYNA, red) was infused either before satiation (A3) or before the probe test (A4). The 15 min interval between satiation and probe test in (A3) was selected to allow for clearance of KYNA prior to the probe test. (B) Intended infusion sites (top) with representative MRIs showing gadolinium contrast after infusion into the MD of two subjects (bottom). (C–J) Histograms indicate means + SEM with individual subject data points overlaid. *=significant difference from control, p<0.05; ^=significantly greater than chance, p<0.05. Full statistical results are presented in *Supplementary file 1c*.

DOI: https://doi.org/10.7554/eLife.37325.004

The following figure supplements are available for figure 2:

**Figure supplement 1.** Low dose KYNA infused before the probe session does not impact reinforcer devaluation.
DOI: https://doi.org/10.7554/eLife.37325.005

**Figure supplement 2.** Magnitude of Disruption in Devaluation Following Inactivation of the MD is similar to that seen after inactivation of either BLA or OFC.
DOI: https://doi.org/10.7554/eLife.37325.006

other regions (*Wellman et al., 2005*; *West et al., 2011*). We microinjected drug or vehicle at one of these two points during the task (see *Figure 2A*), to dissociate potential impairments in value updating (infusion before satiation; *Figure 2A1 and A3*) from impairments in action selection (infusion *prior to probe*; 2A2, 2A4).

Under baseline conditions, animals displayed a slight, but significant, preference for one type of reward (*Figure 2F*). This was evident in their choosing a larger proportion of objects predicting that reward (baseline preference ratio). This pattern is similar to what has been previously reported (*Mitchell et al., 2007*; *West et al., 2011*). Under saline-infused condition, animals displayed robust devaluation following satiation; the devaluation effect was demonstrated by a shift away from choices of objects associated with the devalued food.

Microinjection of MUS either before satiation or before the sated probe test significantly disrupted the devaluation effect compared to sham/saline infusion; this was manifest as a decrease in the proportion shifted (*Figure 2C*). This disruption was evident in all four subjects, resulting in a significant main effect of treatment ($F_{1.1,3.4}$=13.3; p=0.029). Pairwise comparisons revealed a significant impairment in performance when MUS was infused either before satiation (p=0.032) or before the sated probe test (p=0.035). The magnitude of impairment did not differ between these conditions (p=0.13). In contrast to the object probe sessions, MUS injection failed to alter choices in the consummatory probe (*Figure 2D*; $F_{1.3,4.0}$=1.25; p=0.35). Because animals still displayed a typical shift away from sated primary reinforcers in the consummatory probe, the deficits seen in *Figure 2C* cannot be explained by impairment in satiety or valuation of primary reinforcers. Thus, the disruption in the devaluation effect was specific to adjusting the value of the objects to reflect the new value of the primary reinforcer (i.e., the sated food).

In prior studies (*Izquierdo and Murray, 2010*; *Mitchell et al., 2007*; *Wellman et al., 2005*; *West et al., 2011*), it has been reported that after displacing objects and revealing a devalued food reward, monkeys will avoid consuming the devalued food. While we did not systematically record the consumption of food during the object probe tests, anecdotally, there were trials in which the animals did not eat the devalued food after displacing the object.

To determine if deficits in devaluation were secondary to impaired object recognition, we tested animals (on a separate test day) on the concurrent visual discrimination task following drug infusion. MUS injection was associated with a small, but significant impairment in concurrent visual discrimination (*Figure 2E*; t = 4.09, df = 3, p=0.026). It is possible that this finding is due to a drug-related deficit in object recognition, reward associations, and/or appropriate action selection. Interestingly, this deficit also likely contributed to the significant decrease in baseline preference ratio (see Materials and methods) observed after MUS injection (*Figure 2F*; t = 4.28, df = 3, p=0.023). Indeed, unlike under baseline (non-sated) condition when animals typically choose significantly more objects associated with one type of reward over the other, MUS presence abolished this baseline preference ratio (*Figure 2F*).

MUS injection produces long lasting inhibition, with effects evident for hours after drug injection (*Dybdal et al., 2013*). Thus, injection *before* satiation likely results in a suppression of the activity within MD both during satiation and the probe test. Because injections both before and after satiation disrupted devaluation, these data alone were unable to clarify what, if any, role the MD played during the period of selective satiation. To determine if the MD is required specifically during selective satiation, we next turned to microinjection of KYNA, which has a short duration of action (~30 min) (*Forcelli et al., 2014*) and short half-life within the brain (8–30 min) (*Vécsei and Beal, 1990*; *Turski and Schwarcz, 1988*). Based on the timing of our experiments (the probe session was conducted 45 min after drug infusion,~3 half-lives), less than 10% of KYNA is expected to remain during the probe test. Thus, injection *before* satiation is expected to only disrupt activity during the period of satiation and not during the sated probe test. We confirmed, in two monkeys, that lower doses of KYNA infused immediately prior to the sated probe (*Figure 2—figure supplement 1*), did not impact performance. Thus the trace amounts of KYNA remaining when infused prior to satiation are unlikely to be sufficient to impact behavior during the probe test.

KYNA infusion either during satiation, or before the sated probe test, impaired the devaluation effect (*Figure 2G*). This pattern was evident in all three subjects, resulting in a significant main effect ($F_{1.0,2.1}$=52.3; p=0.017) with both test conditions differing significantly from saline-infused control sessions (Infusion before satiation: p=0.021, Infusion before sated probe: p=0.021). Moreover, the

impairment was of greater magnitude when KYNA was infused before the sated probe test (p=0.043).

Similar to MUS infusion, KYNA infusion did not disrupt performance in the consummatory task (*Figure 2H*; $F_{1.1,2.1}$=1.11; p=0.40). In contrast to MUS infusion, KYNA infusion spared performance in concurrent visual discrimination (*Figure 2I*; identical values for all animals under all conditions precluded inferential statistical analysis across treatments). Similarly, KYNA infusion was without effect on baseline preference ratio (*Figure 2J*; t = 0.277, d = 2, p=0.81). In conclusion, impaired devaluation, in the absence of other deficits, indicates that the MD is required both for adjusting the value of object representations and for appropriate action selection.

## Discussion

Our present findings delineate the role of the MD in each of the cognitive processes needed for reinforcer devaluation. We conclude that: [1] Because the expected shift in primary reinforcer preference seen after satiation occurred under all experimental conditions, the MD is not necessary for registering a change in primary reinforcer value. This is consistent with prior studies in which the MD was lesioned. [2] Because inactivation of the MD during satiation impaired reinforcer devaluation, activity in the MD is necessary for adjusting the value of the objects to reflect the new value of primary reinforcers. This deficit is similar to that seen after inactivation of the amygdala or Area 13 of the OFC. [3] Because inactivation of the MD during the probe session (i.e., after satiation was completed) also impaired reinforcer devaluation, activity in the MD is required for appropriate action selection, a role also attributed to Area 11 of the OFC. Thus, we have demonstrated that the MD is required for both reward valuation and action selection; this represents a unique profile within the circuit supporting this behavior, differing from both the amygdala and subregions of the orbitofrontal cortex.

We found a dissociation between the effects of MUS and KYNA with respect to performance on concurrent visual discrimination. While in the case of MUS, at least a portion of the observed deficit in reinforcer devaluation may be due to impaired object discrimination, this is not so for KYNA. After KYNA infusion, concurrent visual discrimination performance was left intact, but deficits in reinforcer devaluation were still evident. In prior studies, large lesions to the MD resulted in deficits in visual recognition/concurrent visual discrimination (*Gaffan and Parker, 2000*; *Aggleton and Mishkin, 1983*) whereas lesions that damaged only the magnocellular portion of the MD (the region we targeted with our drug infusions), did not (*Mitchell et al., 2007*). Because of the differences in duration of action and timing of experimental manipulations, MUS likely spread to a larger area of the MD than did KYNA. While gadolinium is a reasonable proxy of drug spread, and prior functional data from our labs (*Dybdal et al., 2013*; *Malkova et al., 2015*) and others help to estimate the volume of drug spread (*Martin and Ghez, 1999*; *Martin, 1991*; *Allen et al., 2008*), it is indeed a technical limitation of temporary pharmacological inactivation that we cannot directly document the spread of drug for each individual infusion. While speculative, broader inactivation of the MD with MUS may have caused the deficit in concurrent visual discrimination. The degree to which inactivation of the MD would impair performance in other tasks where lesions have produced deficits (*Browning et al., 2015*; *Chakraborty et al., 2016*) remains to be explored.

Inactivation of the MD during the probe test (KYNA infusion after satiation) resulted in a larger deficit than inactivation during satiation; we consider at least two explanations. [1] Amygdala neurons (necessary during satiation) project directly to the OFC (*Timbie and Barbas, 2015*; *Timbie and Barbas, 2014*), and indirectly to the OFC via the MD (*Timbie and Barbas, 2015*; *Ray and Price, 1993*). Thus, even in the absence of MD function, amygdalofrontal projections may partially support updating of the object values. [2] Because Areas 11 and 13 of the OFC support different components of this task, cross-talk between these regions may be critical. Both of these regions are reciprocally interconnected with the MD (*Ray and Price, 1993*), thus, the MD may modulate and facilitate transmission between these cortical regions (see *Figure 2—figure supplement 2* for a comparison across studies). Consistent with this notion, crossed lesions to the OFC and the MD impaired devaluation (*Browning et al., 2015*). Moreover, recent studies in rodents have shown that optogenetic activation of the MD enhances cortico-cortical communication and improves performance on prefrontal cortex dependent tasks, whereas optogenetic silencing of the MD increases errors during prefrontal-dependent task performance (*Schmitt et al., 2017*; *Bolkan et al., 2017*).

Our present findings, which provide novel temporal information regarding the role of MD in reinforcer devaluation, are in general agreement with prior studies. Lesions to the MD produced similar impairments to those that we observed. While the effect magnitude has varied slightly from study to study, the general pattern has been consistent. Given the similarity in the findings between lesion and pharmacological inactivation methodologies, neither compensatory circuit alterations after lesions, nor off-target effects after drug infusion are likely to account for the deficits observed. Thus, together, these data and those previously published underscore a critical role for processing within the MD in behavioral flexibility.

The present data delineate the role of the MD in each of the processes integral to reinforcer devaluation: [1] The MD is not necessary for registering a change in primary reinforcer value. [2] Activity in the MD is necessary for adjusting the value of the objects to reflect the new value of primary reinforcers. [3] The MD is required for appropriate action selection. Together, this pattern of deficits seen with inactivation of the MD is unique; unlike the amygdala, or individual subregions of OFC, the MD is required for multiple components of task performance. Rather than serving as a parallel pathway for information transfer between the amygdala and the OFC, these data instead suggest that the MD functions as a privileged and critical interface between other nodes of the devaluation circuitry.

## Materials and methods

### Animals
Four adult, male, rhesus macaque (Macaca mulatta) were subjects in the present study. At the start of the first reinforcer devaluation study they weighed 8.2–9.8 kg. They were housed with visual access to conspecifics in standard home cages (61 × 74×76 cm each).

Water was available *ad libitum* in the home cage. Meals (LabDiet #5049) were provided twice daily and supplemented with fresh fruit. The first meal was always given after behavioral testing occurred. The study was conducted under a protocol approved by the Georgetown University Animal Care and Use Committee (#2016–1115) and in accordance with the Guide for Care and Use of Laboratory Animals (*Committee for the Update of the Guide for the Care and Use of Laboratory Animals, Institute for Laboratory Animal Research, Division on Earth and Life Studies, National Research Council, 2010*).

These animals were previously tested on a within-session concurrent discrimination learning task (unpublished) and a previous study using the current reinforcer devaluation task with systemic drug administration (*Waguespack et al., 2018*).

### Apparatus and materials
The monkeys were trained in a Wisconsin General Testing Apparatus located in a darkened sound-shielded room. The test tray, which was located at the level of the floor of the monkey transport cage, contained two food wells spaced 18 cm apart (center to center) on the midline of the tray. The wells were 25 mm in diameter and 5 mm deep. The stimuli were 80 junk objects that differed widely in shape, size, color, and texture.

### Food reinforcers
We selected two highly palatable foods that are commonly used as reinforcers across laboratories for this task: fruit snacks and peanuts (*Málková et al., 1997*; *Izquierdo and Murray, 2010*; *Wellman et al., 2005*). Food one was a fruit snack (Sharkbites; General Mills) and food two was half of a honey roasted peanut (Planters; Kraft Foods). Animals did not receive these food reinforcers outside of the context of this task (e.g., as enrichment or reinforcers for other tasks).

### Behavioral training and testing
Monkeys were trained on the task, as described previously (*Málková et al., 1997*).

### Concurrent visual discrimination
The animals were first trained on a set of 40 object-discrimination problems. The objects were placed over the food wells; the monkey could only see and retrieve the food by displacing an object.

In each of the 40 object pairs, one object (S+) was baited with a food reinforcer and the other was unbaited (S−). Half of the S+ objects were baited with fruit snacks and the other half baited with peanuts, intermixed within a session. The S+ and S− assignment of the objects, the order of the object pairs, and the food reinforcer paired with particular objects remained constant across days; however, the left–right positions of the S+ object was pseudorandomized. The intertrial interval between the presentations of two consecutive pairs was 20 s. The monkeys were trained at a rate of one session per day, 5 d a week, until they reached criterion, which was set at a mean of 90% correct responses or better over five consecutive days (i.e., 180 or more correct responses of 200). During the course of this training animals learned, incidentally, the food-object association.

## Probe sessions

Upon reaching criterion on the concurrent visual discrimination, the animals' choices of objects were assessed in probe sessions, in which only baited objects (S+) were used. For each probe session, the 40 S+ objects were randomly assigned to create 20 pairs of probe trials, each offering a choice between an object baited with fruit snack and an object baited with peanut. The left-right position of the object-food pairs was randomized within a session and across days.

On each probe trial, the animal was allowed to select one of the two objects and retrieve the food reward. By repeating this for each of the 20 pairs of probe trials, this allowed for the assessment of baseline preference (*Figure 2F* and *Figure 2J*) for object-food pairs. Animals typically choose more objects baited with one type of food reward (e.g., Food 1) over those baited with the other type (e.g., Food 2). Those chosen in higher numbers are considered 'preferred'. Baseline preference ratio was calculated with the following equation:

$$F_{preferred}/F_{total}$$

where the number of objects chosen with preferred primary reinforcer were divided by the total number of objects chosen.

Other probe sessions (sated probe; see below) were preceded by a devaluation procedure, in which the monkey was allowed to consume one of the two food rewards to satiation (selective satiation).

## Selective satiation

To experimentally induce 'devaluation' of one reinforcer, animals were subjected to selective satiation. Selective satiation sessions were conducted >14 hr after the last feeding. During the satiation procedure, a food box attached to the monkey's home cage was filled with one of the two food reinforcers (either food 1 or food 2) of a measured quantity. The monkey was allowed to eat the food for 15 min, at which point the experimenter re-entered the room and checked the amount of food eaten. If the monkey consumed all of the initially offered food, the experimenter gave the monkey more food. This was repeated until the animal did not take additional food over a five minute period. At that time, the remaining food was removed and measured. In all cases, 30 min was a sufficient time to complete this procedure. After selective satiation, animals were tested in the probe sessions as described above.

Typically-responding animals will shift behavior such that objects associated with the sated (devalued) food are rejected in favor of the objects associated with the non-sated (non-devalued) food. This results in a clear shift in the preference score toward the objects associated with the non-devalued food. Each object was presented only once during the probe session, so that the assessment was uncontaminated by new learning. Therefore, the rejection of an object associated with the sated food is an indication that the devaluation of the food was cognitively transferred to the object.

In a counterbalanced manner, this procedure was repeated for each of the two foods for each test condition. Data across these two devaluation sessions for each condition were used to evaluate the magnitude of devaluation, which was assessed by measuring the proportion shifted, previously described (*Murray et al., 2015*). Proportion shifted was calculated according to the formula:

$$Proportion\,shifted = [(F1_N - F1_D) + [(F2_N - F2_D)]/(F1_N - F2_N)$$

where $F1_N$ is the number of objects baited with fruit snacks chosen during the baseline probe, $F1_D$ is the number objects baited with fruit snack chosen during the sated probe, $F2_N$ is the number of

objects baited with peanuts chosen during the baseline probe and $F2_D$ is the number of objects baited with peanuts chosen during the sated probe.

The amount of food consumed during the selective satiation experiments is shown in *Supplementary file 1a*.

## Cranial infusion platform

Prior to training on the task, each monkey was implanted with a head post and a stereotaxically positioned infusion chamber. The chamber allowed a removable injector, fitted with an infusion cannula of adjustable length to be inserted into predetermined sites in the brain through the guiding channels of a grid. The rectangular chamber (38 mm length x 51 mm width) was covered with a removable top that was secured with four screws. For drug infusions, the top was removed and a grid was inserted to provide guiding channels for the placement of the injector and cannula. The grid (25 mm length, 41 mm width) contained 2 mm - length guiding channels set 1 mm apart (center to center). Surgery was performed as previously described (*Murray et al., 2015*).

A custom-built telescoping injector made of polyethylene terephthalate polyester (Elmeco Engineering) was designed to fit snugly into the infusion grid and allowed for sub-millimeter adjustment of infusion cannula (27 ga stainless steel tubing) length (*Wellman et al., 2005*; *West et al., 2011*). Thus, our final spatial resolution for infusion targeting was 2 mm in the anteroposterior and mediolateral planes, and 0.25 mm in the dorsal-ventral plane.

## Magnetic resonance imaging

Postoperatively, each monkey received one or more T1-weighted scans to determine and/or verify the coordinates for the infusion sites. Scans were conducted as previously described (*Wellman et al., 2005*; *West et al., 2011*) with an effective resolution of $0.25 \times 0.25 \times 0.25$ mm. To calculate infusion site coordinates, a dilute gadolinium solution was instilled into the chamber. Gadolinium contrast allowed for the detection of the position of each of the guide channels within the infusion grid. To verify the volume of diffusion of the infused solution, we infused 1 ul (5 nmol) of an MRI contrast agent, gadolinium (5 mM solution diluted in sterile saline; Magnevist) as previously described. The resolution of this scan was 1 mm x 1 mm x 1 mm. The range of diffusion visualized in MRI sections was limited to a diameter of 3 mm at 60 min after infusion, in agreement with previous gadolinium imaging in our laboratory and others (*Forcelli et al., 2014*; *Krauze et al., 2005*; *Fiandaca et al., 2009*).

## Experimental procedure

For each drug, we evaluated performance in the: (Málková et al., 1997) Object Probe [baseline probe, sated probe], (2) Consummatory Probe, (Timbie and Barbas, 2015) Baseline Preference, and (4) Concurrent Visual Discrimination. For Object and Consummatory Probe tests, we compared performance after drug infusion with matched saline (or sham) infusions. For baseline preference and concurrent visual discrimination, performance was compared to non-injected baselines.

In order to minimize penetrations of the brain, animals received one saline infusion and one sham infusion for each drug. Sham infusions were performed in the same manner as drug infusions except no cannula was inserted. A pair of tests, comprised of one sham infusion and one saline infusion, were performed for each drug (i.e., one pair was performed for MUS, one pair for KYNA). Each pair contained a manipulation performed before satiation and a manipulation performed before the probe session. Furthermore, each pair contained one control with each of the two foods. These tests were performed in a balanced manner, as shown in *Supplementary file 1b*.

Animals AN, TA, EL, and LO were used for experiments with MUS. LO was not included for experiments with KYNA, as he became uncooperative after the completion of experiments with MUS.

## Drug preparation and dose selection

The GABAa receptor agonist muscimol (MUS; Sigma Aldrich) was dissolved in sterile saline at a concentration of 9 mM. The broad-spectrum ionotropic glutamate receptor antagonist, kynurenic acid (KYNA; Sigma Aldrich) was dissolved first in a small quantity of NaOH, neutralized with dilute HCl,

and adjusted to the final concentration of 300 mM by addition of sterile water. All drug solutions were filter sterilized (0.20 μm pore size, Corning) before being stored in 1 ml frozen aliquots (−20C).

Drug doses and concentrations were selected on the basis of our prior microinfusion studies in macaques. We have previously reported functional effects of muscimol within the range of 4.5 to 9 nmol per infusion (i.e., 0.5 to 1 ul of a 9 mM solution) (*Wellman et al., 2005*; *West et al., 2011*). Similarly, we have reported functional effects of KYNA infusion in the range of 300–600 nmol per infusion (*Forcelli et al., 2014*; *Malkova et al., 2015*) (i.e., 1–2 ul of a 300 mM solution).

## Timing of drug infusions relative to behavioral testing

As shown in *Figure 2A*, for MUS, we matched the time between infusion and probe test to ensure an equal diffusion of drug between conditions. To accomplish this, we inserted a 30 min delay between infusion and probe test, when the drug was infused after satiation but before the probe. This strategy is similar to that used in the amygdala and orbitofrontal cortex (*Wellman et al., 2005*; *West et al., 2011*). The duration of action of MUS when microinjected into the brain easily exceeds 90 min; in a prior study, we found that effects of MUS infusion increased over the duration of a 90 min infusion period and observed behavioral responses that in some cases lasted for several hours (*Dybdal et al., 2013*). Thus, it was not possible to allow sufficient time for clearance of MUS prior to the probe test.

In order to determine, what, if any, effect the MD plays exclusively during the period of satiation, we required a compound that would be cleared by the time the probe test occurred. For this reason, we turned to KYNA, which is quickly cleared in the brain and has a relatively short duration of action (*Forcelli et al., 2014*; *Vécsei and Beal, 1990*; *Turski and Schwarcz, 1988*). In rodent studies, the half-life of KYNA in brain has been reported to range from 5 to 30 min (*Turski and Schwarcz, 1988*). Consistent with this timecourse, in a prior study, we observed normalization of cognitive function within 45 min of infusion of KYNA (*Forcelli et al., 2014*) (and unpublished observations) in the primate brain. To maximize time for clearance of KYNA when infused prior to satiation, we inserted a 15 min delay following the satiation procedure; thus the total time from the end of the infusion to the start of the probe was ~45 min. Finally, we infused KYNA at a lower dose (150 nmol) in two animals (EL/LO). This dose of KYNA was without effect on the proportion shifted when infused before the probe test (0.71 and 0.72 after KYNA, as compared to 0.73 and 0.85 after control, respectively; see *Figure 2—figure supplement 1*). Given the half-life of KYNA in brain, even if sparse amounts of KYNA remained when animals were infused before satiation, the concentration of KYNA remaining would not be sufficient to produce the significant deficits observed.

## Intracerebral drug infusions

Drug infusions were performed using an aseptic technique while the monkey was seated in a primate chair with its head posted. The chamber was cleaned immediately prior to each infusion. Coordinates for drug infusion were determined based on individual MRI scans for each monkey. Animals were infused, bilaterally, with either MUS, KYNA, or sterile saline (0.9% NaCl). Drugs were infused at a rate of 0.2 ul/min using the removable injector described above, connected by sterile tubing to Hamilton syringe driven by an injection pump (New Era Pumps Systems). Infusion progress was monitored by the displacement of a small air bubble introduced into the tubing.

## Effect of drug infusion on object probe

Weekly testing occurred per the following schedule (see *Figure 1*). On Day 1, animals were tested on concurrent visual discrimination. If animals correctly chose >90% baited S+ objects during concurrent visual discrimination, on Day two they were tested on a baseline probe. If animals performed below this criterion, they were re-tested on concurrent visual discrimination until they again met the criterion of >90%. On Day 3, animals were again tested on concurrent visual discrimination. On Day 4, animals were infused with drug or saline before satiation or after satiation period, then tested on the object probe session.

## Effect of drug infusion on consummatory probe

To determine if drug manipulations impaired the typical change in value of primary reinforcers that occurs after satiation, we evaluated performance in a consummatory probe session. Immediately

after the completion of both the baseline and sated object probe sessions (described above), animals were presented with 20 pairs of primary reinforcers (fruit snack and peanut) in the absence of objects. The left-right positions of each reinforcer were pseudorandomized across the 20 trials.

### Effect of drug infusion on baseline preference

Animals typically display a small, but significant preference for objects associated with one of the two food reinforcers. To determine if inactivation of the MD impacted this baseline preference, animals were tested in a weekly sequence consisting of concurrent visual discrimination on Day 1, baseline probe on Day 2, and re-tested on the baseline probe following drug infusion on Day 3.

After infusion of KYNA, animals were immediately transferred into the WGTA; following MUS infusion, animals were returned to the home cage for 30 min prior to transfer to the WGTA and testing. This timing was selected to match the timing used for the sated object probe.

### Effect of drug infusion on concurrent visual discrimination

To determine if drug manipulations impaired concurrent visual discrimination performance, animals were tested in a sequence consisting of the concurrent visual discrimination task on Day 1. On Day 2, animals were infused with either MUS or KYNA, transferred to the WGTA, and re-tested on the concurrent visual discrimination task. The timing of drug infusions and testing follows that described above for Baseline Preference testing.

### Statistical analyses

All statistical comparisons were made on a within-subject basis. Results of saline/sham infusions for each animal (both before and after selective satiation) were pooled to generate a cumulative preference shift; in this way, both foods were represented in the control condition.

Devaluation magnitude (proportion shifted) for both the object and consummatory probes were analyzed by analysis of variance with treatment as a within subject condition. The Greenhouse-Geisser correction for violations of sphericity was applied. *A priori* determined pairwise comparisons (Control >Drug Infused) were analyzed using a one-tailed Holm-Sidak corrected post-test. A one-tailed analysis was selected for these comparisons because of our strong *a priori* hypothesis that drug infusion would *impair* not *improve* reinforcer devaluation. We had no *a priori* hypothesis regarding the magnitude of devaluation when drug was infused before satiation as compared to before the probe test, thus these data were analyzed using a two-tailed Holm-Sidak corrected post-test.

Concurrent visual discrimination and baseline preference ratio were analyzed by paired t-test. Baseline preference ratios for MUS also analyzed using a one-sample t-test comparing preference to a neutral preference score (0.5). This analysis was not performed for KYNA infusion, as there were insufficient data points to analyze due to the attrition of one subject prior to KYNA testing.

All data were analyzed using GraphPad Prism (ver 7, Graph Pad, La Jolla, CA). In all cases, *P* values less than 0.05 were considered statistically significant. Full statistical results are shown in *Supplementary file 1c*.

### Data availability statement

The data generated and analyzed during this study are all presented in summary form in the manuscript. Data for object selection during the testing sessions are shown in *Supplementary file 1d*.

## Acknowledgements

We thank Hannah Waguespack and Dr. Victor Santos for technical assistance and Drs. Elizabeth West and Lauren Orefice for helpful comments on prior drafts of this manuscript. This work was funded in part by KL2TR001432 to PAF, work on the manuscript was supported in part by R01099505 to LM, JT was supported by the Intramural Research Program at the National Institute of Mental Health.

## Additional information

### Funding

| Funder | Grant reference number | Author |
|---|---|---|
| National Center for Advancing Translational Sciences | KL2TR001432 | Patrick Alexander Forcelli |
| National Institute of Mental Health | R01MH099505 | Ludise Malkova |

The funders had no role in study design, data collection and interpretation, or the decision to submit the work for publication.

### Author contributions

Evan Wicker, Formal analysis, Investigation, Writing—original draft, Writing—review and editing; Janita Turchi, Resources, Methodology, Writing—review and editing; Ludise Malkova, Conceptualization, Supervision, Funding acquisition, Writing—review and editing; Patrick A Forcelli, Conceptualization, Data curation, Formal analysis, Supervision, Funding acquisition, Writing—review and editing

### Author ORCIDs

Patrick A Forcelli (iD) http://orcid.org/0000-0003-1763-060X

### Ethics

Animal experimentation: The study was conducted under a protocol approved by the Georgetown University Animal Care and Use Committee (#2016-1115) and in accordance with the Guide for Care and Use of Laboratory Animals (26).

### Decision letter and Author response

Decision letter https://doi.org/10.7554/eLife.37325.010
Author response https://doi.org/10.7554/eLife.37325.011

## Additional files

### Supplementary files

• Supplementary file 1. (A) Amount of food consumed (in grams) during selective satiation does not differ across session types. Table shows the amount of food consumed for each animal on each session type. For infusions performed before satiation, mixed effects analysis revealed no significant main effects of food type ($F_{1,15}$=2.6, p=0.13) or treatment ($F_{0.3,2.1}$=0.87, p=0.30) nor a food-by-treatment interaction ($F_{1.9,14}$=0.34, p=0.71). Similarly, for either infusions performed before the probe test, no significant main effects of food type ($F_{1,15}$=1.15, p=0.30) or treatment ($F_{1.4,10.5}$=0.005, p=0.99) nor a food-by-treatment interaction ($F_{0.3,2.4}$=0.07, p=0.52) were detected. Thus, differences in the degree of satiation across session types cannot contribute to the deficits we report. (B) Testing Order for Control Infusions. As described in the methods, to minimize penetrations of the brain, animals received one saline infusion and one sham infusion for each drug. Sham infusions were performed in the same manner as drug infusions except no cannula was inserted. A pair of tests, comprised of one sham infusion and one saline infusion, were performed for each drug (i.e., one pair was performed for MUS, one pair for KYNA). Each pair contained a manipulation performed before satiation and a manipulation performed before the probe session. Furthermore, each pair contained one control with each of the two foods. In the Table, SHAM indicates a sham infusion, SAL indicates a saline infusion. Boxes shaded blue are control infusions for MUS, while those shaded red are control infusions for KYNA. One animal (AN), received an additional control session for KYNA. Data across these two control sessions were summed for analysis. Since LO was not included in the KYNA experiments, this animal only received two control infusions (i.e., those for MUS). (C) Statistical Results for *Figure 2*. The table shows detailed statistical results for the data presented in *Figure 2*. Post tests (where appropriate) were Holm-Sidak corrected for multiple comparisons. (D) Tabulated

number of objects associated with each food chosen during the non-sated probe (F$_N$) and during the sated probe (F$_D$). The saline infusions followed the schedule shown in *Supplementary file 1a*. *=summed across two control sessions. N/A = not tested.

DOI: https://doi.org/10.7554/eLife.37325.007

• Transparent reporting form
DOI: https://doi.org/10.7554/eLife.37325.008

## Data availability

The data generated and analyzed during this study are all presented in the manuscript. Raw data for object selection during the testing sessions are shown in Supplementary files 1a and d.

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
