## [Decision Letter]

Thank you for submitting your article "Mediodorsal Thalamus Is Required for Discrete Phases of Goal-Directed Behavior in Macaques" for consideration by *eLife*. Your article has been reviewed by two peer reviewers, one of whom, Geoffrey Schoenbaum is a member of our Board of Reviewing Editors and the evaluation has been overseen by Sabine Kastner as the Senior Editor.

The reviewers have discussed the reviews with one another and the Reviewing Editor has drafted this decision to help you prepare a revised submission.

Summary:

In this experiment, the authors apply a standard version of a reinforcer devaluation used in primates to explore the role of MD thalamus. Monkeys were trained to discriminate a series of objects associated with two rewards, then they were given choices between objects after satiation on one of the foods. Critically food was delivered in the probe test. MD was inactivated by infusion of either muscimol or kynurenic acid prior to either the satiation or choice test. The authors report that inactivation at either time point, even by the shorter acting kynurenic acid, disrupted the normal effect of satiation on choice behavior. This was despite no effect on the devaluation of the food items and minimal or no effects on object discrimination and preference ratios. The authors conclude that MD is therefore necessary for both updating of the secondary reinforcers value as well as the use of that information to guide choices.

Overall the reviewers thought the study was excellent. The rationale was clearly presented, and the experiment was appropriate to address it. The authors did a great job of presenting the results in a clear and easy to understand manner. Generally, there were no major problems with the conclusions, which fit well with extant data, while extending it in potentially important ways.

Essential revisions:

There were only a few revisions judged to be essential. They are listed in the reviews but are emphasized here for clarity. One was to avoid the use of the term secondary reinforcer, since this is not formally shown. It was also felt that a deeper consideration of some caveats related to pharmacological inactivation in monkeys might be useful, and some additional discussion of the other data on the role of MD in the Discussion section.

Reviewer #1:

In this experiment, the authors apply a standard version of a reinforcer devaluation used in primates to explore the role of MD thalamus. Monkeys were trained to discriminate a series of objects associated with two rewards, then they were given choices between objects after satiation on one of the foods. Critically food was delivered in the probe test. MD was inactivated by infusion of either muscimol or kynurenic acid prior to either the satiation or choice test. The authors report that inactivation at either time point, even by the shorter acting kynurenic acid, disrupted the normal effect of satiation on choice behavior. This was despite no effect on the devaluation of the food items and minimal or no effects on object discrimination and preference ratios. The authors conclude that MD is therefore necessary for both updating of the secondary reinforcers value as well as the use of that information to guide choices.

Overall, I thought the study was excellent. The rationale was clearly presented, and the experiment was appropriate to address it. The authors did a great job of presenting the results in a clear and easy to understand manner. And I do not have major problems with their conclusions. It seems to me that they fit well with extant data, while extending it in potentially important ways.

I do have a couple of specific requests however.

1) Can the authors clarify that satiety affected consumption of the foods in the actual test? Since they were given, did the monkeys eat them? Was there any evidence that satiety affected this (actual consumption, latency?) when the monkeys did pick the object associated with the satiated food? And did the inactivation affect this?

2) Can the authors spend some additional time in the Discussion section on the prior studies? It seems to me that this is important, first since the effects here basically replicate that work and second, I think it mitigates concerns that acutely disrupting MD is having non-specific effects on these other areas. With acute inactivation, I would otherwise be concerned that shutting off such a huge source of inputs to OFC was having non-specific effects. That similar effects are obtained with lesions of MD (or crossed lesions) where compensation for this can occur I think bolsters the case that this is a specific effect – i.e. due to real processing in MD.

Reviewer #2:

This paper interrogates the involvement of the mediodorsal thalamus (MD) in a test of goal-directed behavior, the reinforcer devaluation task, in rhesus monkeys. The authors use site-specific drug infusions targeted at the MD in order to test the relative involvement of this structure in adjusting the representation of the reinforcer after satiation, versus adapting choice behavior in response to the current value of the reward. These procedures have revealed different roles in choice behavior for amygdala and for subdivisions of orbital prefrontal cortex, but the MD has not been tested in this way. The authors find that MD is not necessary for satiation per se or adjusting behavior towards the rewards themselves but is involved both in updating the representation of the reward after satiation and for adjusting choice behavior based on the updated value of the reward (i.e., appropriate goal-directed behavior).

These are challenging experiments to carry out in monkeys and the findings provide new information about the involvement of the MD in this circuitry. Given the interest in neuromodulation of MD as a strategy for normalizing deficits in prefrontal function, these findings are an important piece of the puzzle of understanding how MD and prefrontal cortex interact in the service of behavior.

I find the characterization of the objects the monkeys are choosing as "secondary reinforcers" (Introduction, Results section and Discussion section) confusing. Although stimuli paired with reward may acquire reinforcing power, they need not do so. Indeed, the presence of devaluation effects in the object discrimination task implies that behavior is being guided by the representation of the reward rather than reinforcing power acquired by the objects themselves.

The authors might consider discussing some of the limitations of this technique, including precisely delineating the targeting and spread of the drug infusions. This is a challenge with all temporary inactivation methods. Some of the gadolinium infusions appear a bit laterally placed within the MD in Figure 2; the authors also speculate that the difference in findings on object discrimination and baseline choice between muscimol and kynurenic acid are due to differences in extent of diffusion and duration of action, but there is no direct investigation of this (although it would be very challenging to do so). The cleaner effects of the kynurenate infusions clearly support the authors' conclusions, so I have no problems with the inferences that are drawn, but I think the methodological limitations should be more explicitly addressed.

---

## [Author Response]

Essential revisions:There were only a few revisions judged to be essential. They are listed in the reviews but are emphasized here for clarity. One was to avoid the use of the term secondary reinforcer, since this is not formally shown. It was also felt that a deeper consideration of some caveats related to pharmacological inactivation in monkeys might be useful, and some additional discussion of the other data on the role of MD in the Discussion section.Reviewer #1:1) Can the authors clarify that satiety affected consumption of the foods in the actual test? Since they were given, did the monkeys eat them? Was there any evidence that satiety affected this (actual consumption, latency?) when the monkeys did pick the object associated with the satiated food? And did the inactivation affect this?

Unfortunately, we did not systematically record the consumption of food during testing. Anecdotally, there were some trials in which the animals did not eat the devalued food after displacing the object. This has been noted in the text (Results section).

2) Can the authors spend some additional time in the Discussion section on the prior studies? It seems to me that this is important, first since the effects here basically replicate that work and second, I think it mitigates concerns that acutely disrupting MD is having non-specific effects on these other areas. With acute inactivation, I would otherwise be concerned that shutting off such a huge source of inputs to OFC was having non-specific effects. That similar effects are obtained with lesions of MD (or crossed lesions) where compensation for this can occur I think bolsters the case that this is a specific effect – i.e. due to real processing in MD.

We added a paragraph to the Discussion section on this point.

Reviewer #2:I find the characterization of the objects the monkeys are choosing as "secondary reinforcers" (Introduction, Results section and Discussion section) confusing. Although stimuli paired with reward may acquire reinforcing power, they need not do so. Indeed, the presence of devaluation effects in the object discrimination task implies that behavior is being guided by the representation of the reward rather than reinforcing power acquired by the objects themselves.

When discussing our data, we have changed secondary reinforcer. However, we kept secondary reinforcer as part of the broader concept of reinforcer devaluation in the Abstract and Introduction. We also kept this in the impact statement, as there is not a precise and concise other way of saying this.

The authors might consider discussing some of the limitations of this technique, including precisely delineating the targeting and spread of the drug infusions. This is a challenge with all temporary inactivation methods. Some of the gadolinium infusions appear a bit laterally placed within the MD in Figure 2; the authors also speculate that the difference in findings on object discrimination and baseline choice between muscimol and kynurenic acid are due to differences in extent of diffusion and duration of action, but there is no direct investigation of this (although it would be very challenging to do so). The cleaner effects of the kynurenate infusions clearly support the authors' conclusions, so I have no problems with the inferences that are drawn, but I think the methodological limitations should be more explicitly addressed.

We have added an additional discussion of this and made our conclusions regarding drug spread a bit more circumspect (Discussion section).